# Latency-aware Spatial-wise Dynamic Networks

**Yizeng Han**[1][*] **Zhihang Yuan**[2][*] **Yifan Pu**[1][*] **Chenhao Xue**[2]
**Shiji Song**[1] **Guangyu Sun**[2] **Gao Huang**[1][†]
[1] Department of Automation, BNRist, Tsinghua University, Beijing, China
[2] School of Electronics Engineering and Computer Science, Peking University, Beijing, China
{hanyz18, pyf20}@mails.tsinghua.edu.cn, {shijis, gaohuang}@tsinghua.edu.cn
{yuanzhihang, xch927027, gsun}@pku.edu.cn

## Abstract

Spatial-wise dynamic convolution has become a promising approach to improving the inference efficiency of deep networks. By allocating more computation to the most informative pixels, such an *adaptive* inference paradigm reduces the spatial redundancy in image features and saves a considerable amount of unnecessary computation. However, the *theoretical* efficiency achieved by previous methods can hardly translate into a *realistic* speedup, especially on the multi-core processors (*e.g.* GPUs). The key challenge is that the existing literature has only focused on designing algorithms with minimal *computation*, ignoring the fact that the practical latency can also be influenced by *scheduling strategies* and *hardware properties*. To bridge the gap between theoretical computation and practical efficiency, we propose a *latency-aware* spatial-wise dynamic network (LASNet), which performs *coarse-grained* spatially adaptive inference under the guidance of a novel *latency prediction model*. The latency prediction model can efficiently estimate the inference latency of dynamic networks by simultaneously considering algorithms, scheduling strategies, and hardware properties. We use the latency predictor to guide both the algorithm design and the scheduling optimization on various hardware platforms. Experiments on image classification, object detection and instance segmentation demonstrate that the proposed framework significantly improves the practical inference efficiency of deep networks. For example, the average latency of a ResNet-101 on the ImageNet validation set could be reduced by 36% and 46% on a server GPU (Nvidia Tesla-V100) and an edge device (Nvidia Jetson TX2 GPU) respectively without sacrificing the accuracy. Code is available at https://github.com/LeapLabTHU/LASNet.

## 1 Introduction

Dynamic neural networks [7] have attracted great research interests in recent years. Compared to static models [11, 17, 13, 23] which treat different inputs equally during inference, dynamic networks can allocate the computation in a *data-dependent* manner. For example, they can conditionally skip the computation of network layers [15, 9, 32, 30] or convolutional channels [19, 1], or perform *spatially* adaptive inference on the most informative image regions (*e.g.* the foreground areas) [6, 5, 31, 35, 33, 8]. Spatial-wise dynamic networks, which typically decide whether to compute each feature pixel with *masker* modules [5, 31, 35, 8] (Figure 1 (a)), have shown promising results in improving the inference efficiency of convolution neural networks (CNNs).

Despite the remarkable *theoretical* efficiency achieved by spatial-wise dynamic networks [5, 31, 35], researchers have found it challenging to translate the theoretical results into *realistic* speedup,

---

[*]Equal contribution.
[†]Corresponding author.

36th Conference on Neural Information Processing Systems (NeurIPS 2022).

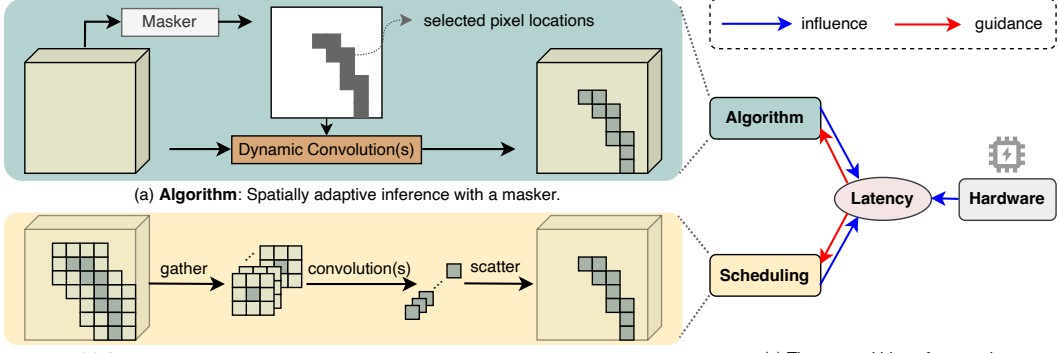

Figure 1: An overview of our method. (a) illustrates the spatially adaptive inference *algorithm*; (b) is the *scheduling* strategy; and (c) presents the three key factors to the practical latency. For a given hardware, the latency is used to *guide* our algorithm design and scheduling optimization.

especially on some multi-core processors, *e.g.,* GPUs [35, 3, 8]. The challenges are two-fold: 1) most previous approaches [5, 31, 35] perform spatially adaptive inference at the finest *granularity*: every *pixel* is flexibly decided whether to be computed or not. Such flexibility induces non-contiguous *memory access* [35] and requires specialized *scheduling strategies* (Figure 1 (b)); 2) the existing literature has only adopted the *hardware-agnostic* FLOPs (floating-point operations) as an inaccurate proxy for the efficiency, lacking latency-aware guidance on the algorithm design. For dynamic networks, the adaptive computation with sub-optimal scheduling strategies further enlarges the discrepancy between the theoretical FLOPs and the practical latency. Note that it has been validated by previous works that the latency on CPUs has a strong correlation with FLOPs [8, 35]. Therefore, we mainly focus on the GPU platform in this paper, which is more challenging and less explored.

We address the above challenges by proposing a *latency-aware* spatial-wise dynamic network (LASNet). Three key factors to the inference latency are considered: the *algorithm*, the *scheduling strategy*, and the *hardware properties*. Given a target hardware device, we directly use the latency, rather than the FLOPs, to *guide* our algorithm design and scheduling optimization (Figure 1 (c)).

Because the memory access pattern and the scheduling strategies in our dynamic operators differ from those in static networks, the libraries developed for static models (*e.g.* cuDNN) are sub-optimal for the acceleration of dynamic models. Without the support of libraries, each dynamic operator requires scheduling optimization, code optimization, compiling, and deployment for each device. Therefore, it is laborious to evaluate the network latency on different hardware platforms. To this end, we propose a novel *latency prediction model* to efficiently estimate the realistic latency of a network by simultaneously considering the aforementioned three factors. Compared to the hardware-agnostic FLOPs, our predicted latency can better reflect the practical efficiency of dynamic models.

Guided by this latency prediction model, we establish our latency-aware spatial-wise dynamic network (LASNet), which adaptively decides whether to allocate computation on feature *patches* instead of *pixels* [5, 31, 35] (Figure 2 top). We name this paradigm as spatially adaptive inference at a *coarse granularity*. While less flexible than the pixel-level adaptive computation in previous works [5, 31, 35], it facilitates more contiguous memory access, benefiting the realistic speedup on hardware. The scheduling strategy and the implementation are further ameliorated for faster inference.

It is worth noting that LASNet is designed as a general framework in two aspects: 1) the coarse-grained spatially adaptive inference paradigm can be conveniently implemented in various CNN backbones, *e.g.,* ResNets [11], DenseNets [17] and RegNets [25]; and 2) the latency predictor is an off-the-shell tool which can be directly used for various computing platforms (*e.g.* server GPUs and edge devices).

We evaluate the performance of LASNet on multiple CNN architectures on image classification, object detection, and instance segmentation tasks. Experiment results show that our LASNet improves the efficiency of deep CNNs both theoretically and practically. For example, the inference latency of ResNet-101 on ImageNet [4] is reduced by 36% and 46% on an Nvidia Tesla V100 GPU and an Nvidia Jetson TX2 GPU, respectively, without sacrificing the accuracy. Moreover, the proposed method outperforms various lightweight networks in a low-FLOPs regime.

Our main contributions are summarized as follows:

1. We propose LASNet, which performs coarse-grained spatially adaptive inference guided by the practical latency instead of the theoretical FLOPs. To the best of our knowledge, LASNet is the first framework that directly considers the real latency in the design phase of dynamic neural networks;

2. We propose a latency prediction model, which can efficiently and accurately estimate the latency of dynamic operators by simultaneously considering the algorithm, the scheduling strategy, and the hardware properties;

3. Experiments on image classification and downstream tasks verify that our proposed LASNet can effectively improve the practical efficiency of different CNN architectures.

## 2 Related works

**Spatial-wise dynamic network** is a common type of dynamic neural networks [7]. Compared to static models which treat different feature locations evenly during inference, these networks perform spatially adaptive inference on the most informative regions (*e.g.*, foregrounds), and reduce the unnecessary computation on less important areas (*e.g.*, backgrounds). Existing works mainly include three levels of dynamic computation: resolution level [36, 37], region level [33] and pixel level [5, 31, 35]. The former two generally manipulate the network inputs [33, 37] or require special architecture design [36]. In contrast, pixel-level dynamic networks can flexibly skip the convolutions on certain feature pixels in arbitrary CNN backbones [5, 31, 35]. Despite its remarkable *theoretical* efficiency, pixel-wise dynamic computation brings considerable difficulty to achieving *realistic* speedup on multi-core processors, *e.g.*, GPUs. Compared to the previous approaches [5, 31, 35] which only focus on reducing the theoretical computation, we propose to directly use the latency to guide our algorithm design and scheduling optimization.

**Hardware-aware network design.** To bridge the gap between theoretical and practical efficiency of deep models, researchers have started to consider the real latency in the network design phase. There are two lines of works in this direction. One directly performs speed tests on targeted devices, and summarizes some guidelines to facilitate *hand-designing* lightweight models [23]. The other line of work *searches* for fast models using the neural architecture search (NAS) technique [29, 34]. However, all existing works try to build *static* models, which have intrinsic computational redundancy by treating different inputs in the same way. However, speed tests for dynamic operators on different hardware devices can be very laborious and impractical. In contrast, our proposed latency prediction model can efficiently estimate the inference latency on any given computing platforms by simultaneously considering algorithm design, scheduling strategies and hardware properties.

## 3 Methodology

In this section, we first introduce the preliminaries of spatially adaptive inference, and then demonstrate the architecture design of our LASNet. The latency prediction model is then explained, which guides the granularity settings and the scheduling optimization for LASNet. We further present the implementation improvements for faster inference, followed by the training strategies.

### 3.1 Preliminaries

**Spatially adaptive inference.** The existing spatial-wise dynamic networks are usually established by attaching a masker $\mathcal{M}$ in each convolutional block of a CNN backbone (Figure 1 (a)). Specifically, let $\mathbf{x} \in \mathbb{R}^{H \times W \times C}$ denote the input of a block, where $H$ and $W$ are the feature height and width, and $C$ is the channel number. The masker $\mathcal{M}$ takes $\mathbf{x}$ as input, and generates a binary-valued spatial mask $\mathbf{M} = \mathcal{M}(\mathbf{x}) \in \{0, 1\}^{H \times W}$. Each element of $\mathbf{M}$ determines whether to perform convolution operations on the corresponding location of the output feature. The unselected regions will be filled with the values from the input [5, 31] or obtained via interpolation [35]. We define the *activation rate* of a block as $r = \frac{\sum_{i,j} \mathbf{M}_{i,j}}{H \times W}$, representing the ratio of the calculated pixels.

**Scheduling strategy.** During inference, the current scheduling strategy for spatial-wise dynamic convolutions generally involve three steps [26] (Figure 1 (b)): 1) *gathering*, which re-organizes the selected pixels (if the convolution kernel size is greater than $1 \times 1$, the neighbors are also required) along the *batch* dimension; 2) *computation*, which performs convolution on the gathered input; and 3) *scattering*, which fills the computed pixels on their corresponding locations of the output feature.

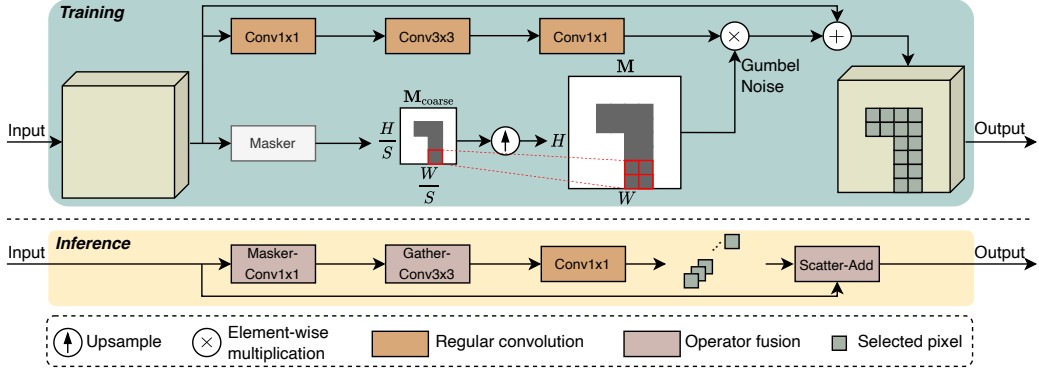

Figure 2: Our proposed LASNet block. Top: we first generate a low-resolution spatial mask $\mathbf{M}_{\text{coarse}}$, which is then upsampled to obtain the mask $\mathbf{M}$ with the same size as the output feature. Gumbel Softmax [18, 24] is used for end-to-end training (Sec. 3.5). Bottom: the scheduling optimization is performed to decrease the memory access for faster inference (Sec. 3.4).

**Limitations.** Compared to performing convolutions on the entire feature map, the aforementioned scheduling strategy reduces the computation while bringing considerable overhead to *memory access* due to the mask generation and the non-contiguous memory access. Such overhead would increase the overall latency, especially when the *granularity* of dynamic convolution is at the finest pixel level.

## 3.2 Architecture design

**Spatial granularity.** As mentioned above, *pixel*-level dynamic convolutions [5, 31, 35] raise substantial challenges to achieving realistic speedup on multi-core processors due to the non-contiguous memory access. To this end, we propose to optimize the *granularity* of spatially adaptive inference. Specifically, take the commonly used bottleneck structure in [11] as an example, our coarse-grained spatial-wise dynamic convolutional block is illustrated in Figure 2. Instead of directly producing a mask with the shape of $H \times W$, we first generate a low-resolution mask $\mathbf{M}_{\text{coarse}} \in \{0, 1\}^{\frac{H}{S} \times \frac{W}{S}}$, where $S$ is named as the *spatial granularity*. Each element in $\mathbf{M}_{\text{coarse}}$ determines the computation of an $S \times S$-sized feature patch. For instance, the feature size in the first ResNet stage[3] is $56 \times 56$. Then the possible choices for $S$ are $\{1, 2, 4, 7, 8, 14, 28, 56\}$. The mask $\mathbf{M}_{\text{coarse}}$ is then upsampled to the size of $H \times W$. Notably, $S = 1$ means that the granularity is still at the pixel level as previous methods [5, 31, 35]. Note that the other extreme situation ($S = 56$) is not considered in this paper, when the masker directly determines whether to skip the whole block (*i.e.* layer skipping [30, 32]). Such an overly aggressive approach will lead to a considerable drop of accuracy, as we presented in Appendix C.2. The masker is composed of a pooling layer followed by a $1 \times 1$ convolution.

**Differences to existing works.** Without using the interpolation operation [35] or the carefully designed two-branch structure [8], the proposed block architecture is simple and sufficiently general to be plugged into most backbones with minimal modification. Our formulation is mostly similar to that in [31], which could be viewed as a variant of our method with the spatial granularity $S$=1 for all blocks. Instead of performing spatially adaptive inference at the finest pixel level, our granularity $S$ is optimized under the guidance of our *latency prediction model* (details are presented in the following Sec. 4.2) to achieve *realistic speedup* on target computing platforms.

## 3.3 Latency prediction model

As stated before, it is laborious to evaluate the latency of dynamic operators on different hardware platforms. To efficiently seek preferable granularity settings on arbitrary hardware devices, we propose a latency prediction model $\mathcal{G}$, which can directly *predict* the delay of executing dynamic operators on any target devices. For a spatial-wise dynamic convolutional block, the latency predictor $\mathcal{G}$ takes the hardware properties $\mathbf{H}$, the layer parameters $\mathbf{P}$, the spatial granularity $S$, and the activation rate $r$ as input and predicts the latency $\ell$ of a dynamic convolutional block: $\ell = \mathcal{G}(\mathbf{H}, \mathbf{P}, S, r)$.

**Hardware modeling.** We model a hardware device as multiple processing engines (PEs), and parallel computation can be executed on these PEs. As shown in Figure 3, we model the memory system as a

---

[3]Here we refer to a stage as the cascading of multiple blocks which process features with the same resolution.

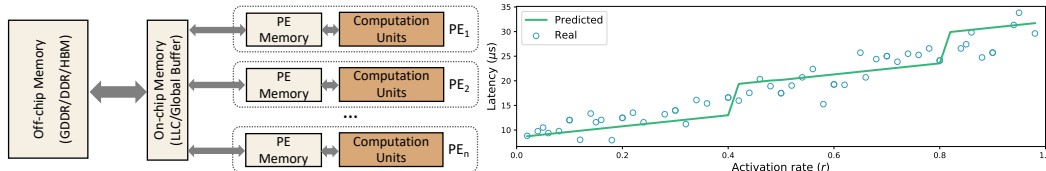

Figure 3: Our hardware model.          Figure 4: Latency prediction results.

three-level structure [12]: 1) off-chip memory, 2) on-chip global memory, and 3) memory in PE. Such a hardware model enables us to accurately predict the cost on both *data movement* and *computation*.

**Latency prediction.** When simulating the data movement procedure, the efficiency of non-contiguous memory accesses under different granularity $S$ settings is considered. As for the computation latency, it is important to adopt a proper scheduling strategy to increase the parallelism of computation. Therefore, we search for the optimal scheduling (the configuration of tiling and in-PE parallel computing) of dynamic operations to maximize the utilization of hardware resources. A more detailed description of our latency prediction model is presented in Appendix A.

**Empirical validation.** We take the first block in ResNet-101 as an example and vary the activation rate $r$ to evaluate the performance of our prediction model. The comparison between our predictions and the real testing latency on the Nvidia V100 GPU is illustrated in Figure 4, from which we can observe that our predictor can accurately estimate the real latency in a wide range of activation rates.

### 3.4 Implementation details

We use general optimization methods like fusing activation functions and batch normalization layers into convolution layers. We also optimize the specific operators in our spatial-wise dynamic convolutional blocks as follows (see also Figure 2 for an overview).

**Fusing the masker and the first convolution.** As mentioned in Sec. 3.1, the masker in each block consumes very little computation, but it takes the whole feature map as input. Therefore, it is a *memory-bounded* operation (the inference time is mainly spent on memory access). Since the masker and the first convolution in the block share the same input, there is an opportunity to fuse these two operations to avoid the repeated access of the input data. Note that a spatial-wise dynamic convolution requires the output of the masker. If we fuse the two layers, the first convolution will be changed to a static operation, which may increase the inference latency. There exists a threshold of activation rate $r_{\text{th}}$, when $r > r_{\text{th}}$, the overall latency can be reduced. We decide whether to fuse them according to the average activation rate. See more details in Appendix B.

**Fusing the gather operation and the dynamic convolution.** Traditional approaches first gather the input pixels of the first dynamic convolution in a block (Figure 1 (b)). The gather operation is also a *memory-bounded* operation. Furthermore, when the size of the convolution kernel exceeds $1 \times 1$, the area of input patches may overlap, resulting in repeated memory load/store. We fuse the gather operation into the dynamic convolution to reduce the memory access.

**Fusing the scatter operation and the add operation.** Traditional approaches scatter the output pixels of the last dynamic convolution, and then execute the element-wise addition (Figure 1 (b)). We fuse these two operators to reduce the memory access. The ablation study in Sec. 4.4 validates the effectiveness of the proposed fusing methods.

### 3.5 Training

**Optimization of non-differentiable maskers.** The masker modules are required to produce binary-valued spatial masks for making discrete decisions, and cannot be directly optimized with back propagation. Following [35, 31, 8], we adopt straight-through Gumbel Softmax [18, 24] to train the network in an end-to-end fashion. Specifically, let $\tilde{\mathbf{M}} \in \mathbb{R}^{H \times W \times 2}$ denote the output of the mask generator. The decisions are obtained with the argmax function during inference. In the training phase, a differentiable approximation is defined by replacing the argmax operation with a Softmax:

$$\hat{\mathbf{M}} = \frac{\exp\left\{\left(\log\left(\tilde{\mathbf{M}}_{:,:,0}\right) + \mathbf{G}_{:,:,0}\right)/\tau\right\}}{\sum_{k=0}^{1}\exp\left\{\left(\log\left(\tilde{\mathbf{M}}_{:,:,k}\right) + \mathbf{G}_{:,:,k}\right)/\tau\right\}} \in [0,1]^{H \times W}, \tag{1}$$

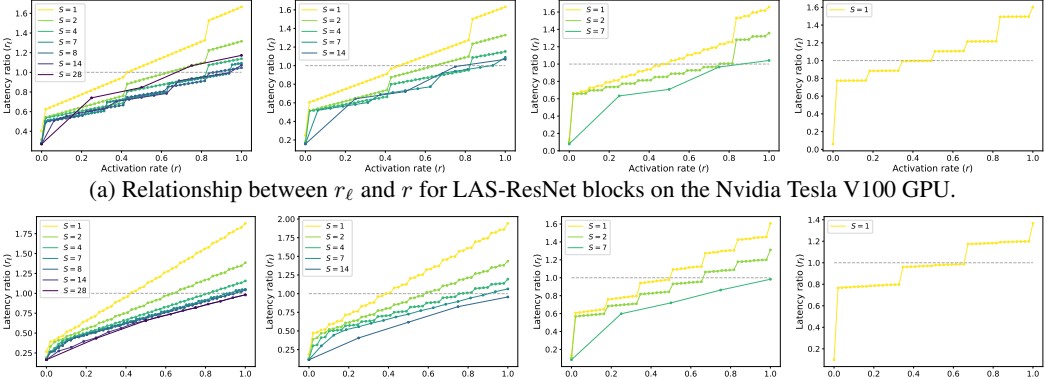

(a) Relationship between $r_\ell$ and $r$ for LAS-ResNet blocks on the Nvidia Tesla V100 GPU.

(b) Relationship between $r_\ell$ and $r$ for LAS-RegNetY-800MF blocks on the Nvidia Jetson TX2 GPU.

Figure 5: Latency prediction results of LAS-ResNet blocks on V100 (a) and LAS-RegNet blocks on TX2 (b). For both networks, we plot the relationship between the latency ratio $r_\ell$ and the activation rate $r$ for the blocks in 4 stages with the convolutional stride 1. The practical efficiency is only improved when $r_\ell < 1$. Note that $S = 1$ can harm the practical latency even for a small $r$ (reduced computation), while a larger $S$ will alleviate this problem. See detailed analysis in Sec. 4.2.

where $\tau$ is the Softmax temperature. Following the common practice [31, 8], we let $\tau$ decrease exponentially from 5.0 to 0.1 in training to facilitate the optimization of maskers.

**Training objective.** The FLOPs of each spatial-wise dynamic convolutional block can be calculated based on our defined activation rate $r$ [31]. Then we can obtain the FLOPs of the overall dynamic network $F_{\mathrm{dyn}}$. Let $F_{\mathrm{stat}}$ denotes the FLOPs of its static counterpart. We optimize their ratio to approximate a target $0 < t < 1$: $L_{\mathrm{FLOPs}} = (\frac{F_{\mathrm{dyn}}}{F_{\mathrm{stat}}} - t)^2$. In addition, we define loss item $L_{\mathrm{bounds}}$ as in [31] to constrain the upper bound and the lower bound of activation rates in early training epochs.

We further propose to leverage the static counterparts of our dynamic networks as "teachers" to guide the optimization procedure. Let $\mathbf{y}$ and $\mathbf{y}'$ denote the output logits of a dynamic model ("student") and its "teacher", respectively. Our final loss can be written as

$$L = L_{\mathrm{task}} + \alpha(L_{\mathrm{FLOPs}} + L_{\mathrm{bounds}}) + \beta T^2 \cdot \mathrm{KL}(\sigma(\mathbf{y}/T)||\sigma(\mathbf{y}'/T)), \qquad (2)$$

where $L_{\mathrm{task}}$ represents the task-related loss, *e.g.*, cross-entropy loss in image classification. $\mathrm{KL}(\cdot||\cdot)$ denotes the Kullback–Leibler divergence, and $\alpha, \beta$ are the coefficients balancing these items. We use $\sigma$ to denote the log-Softmax function, and $T$ is the temperature for computing KL-divergence.

## 4 Experiments

In this section, we first introduce the experiment settings in Sec. 4.1. Then the latency of different granularity settings are analyzed in Sec. 4.2. The performance of our LASNet on ImageNet is further evaluated in Sec. 4.3, followed by the ablation studies in Sec. 4.4. Visualization results are illustrated in Sec. 4.5, and we finally validate our method on the object detection task (Sec. 4.6). The results on the instance segmentation task are presented in . For simplicity, we add "LAS-" as a prefix before model names to denote our LASNet, *e.g.*, LAS-ResNet-50.

### 4.1 Experiment settings

**Latency prediction.** Various types of hardware platforms are tested, including a server GPU (Tesla V100), a desktop GPU (GTX1080) and edge devices (*e.g.*, Nvidia Nano and Jetson TX2). The major properties considered by our latency prediction model include the number of processing engines (#PE), the floating-point computation in a processing engine (#FP32), the frequency and the bandwidth. It can be observed that the server GPUs generally have a larger #PE than the IoT devices. The batch size is set as 1 for all dynamic models and computing platforms.

**Image classification.** The image classification experiments are conducted on the ImageNet [4] dataset. Following [31], we initialize the backbone parameter from a pre-trained checkpoint[4], and

---

[4]We use the torchvision pre-trained models at `https://pytorch.org/vision/stable/models.html`.

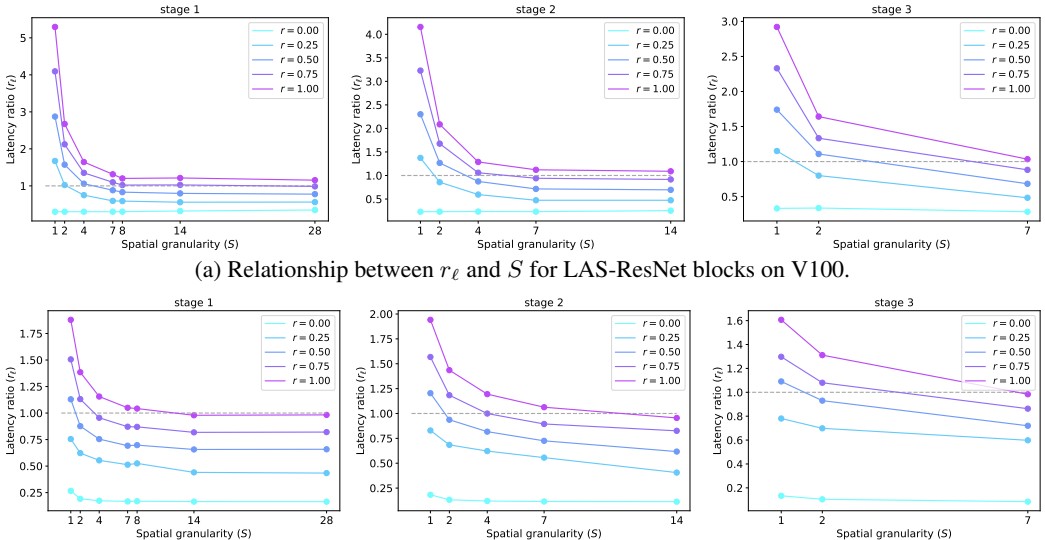

(a) Relationship between $r_\ell$ and $S$ for LAS-ResNet blocks on V100.

(b) Relationship between $r_\ell$ and $S$ for LAS-RegNetY-800MF blocks on Nvidia Jetson TX2 GPU.

Figure 6: The relationship between the latency ratio $r_\ell$ and the spatial granularity $S$.

finetune the whole network for 100 epochs with the loss function in Eq. (2). We fix $\alpha = 10, \beta = 0.5$ and $T = 4.0$ for all dynamic models. More details are provided in Appendix B.

## 4.2 Latency prediction results

In this subsection, we present the latency prediction results of the spatial-wise dynamic convolutional blocks in two different models: LAS-ResNet-101 [11] (on V100) and LAS-RegNetY-800MF [25] (on TX2). All the blocks have the bottleneck structure with different channel numbers and convolution groups, and the RegNetY is equipped with Squeeze-and-Excitation (SE) [14] modules.

We first define $\ell_{\text{dyn}}$ as the latency of a spatial-wise dynamic convolutional block, and $\ell_{\text{stat}}$ as that of a static block without a masker. Their ratio is denoted as $r_\ell = \frac{\ell_{\text{dyn}}}{\ell_{\text{stat}}}$. We investigate the relationship between $r_\ell$ and the activation rate $r$ (cf. Sec. 3.5) for different *granularity* settings. The results in Figure 5 demonstrate that: 1) even equipped with our special optimization on the scheduling strategies, pixel-level spatially adaptive inference ($S$=1) *cannot always* improve the practical efficiency. Such fine-grained adaptive inference is adopted by most previous works [31, 35], and our result can explain the reason why they can only achieve realistic speedup on less powerful CPUs [35] or specialized devices [3]; 2) a proper granularity $S > 1$ effectively alleviates this problem on both hardware devices. By setting $S > 1$, realistic speedup could be achieved with larger activation rates.

The latency prediction results are further used to seek for a preferable granularity setting for the first 3 stages (we fix $S = 1$ for the last stage, where the feature resolution is $7 \times 7$). Therefore, we plot the relationship between $r_\ell$ and $S$ in Figure 10. It can be observed that: 1) $r_\ell$ generally decreases with $S$ increasing for a given $r$; 2) an overly large $S$ (less flexible adaptive inference) brings insignificant improvement on both devices. Especially, enlarging $S$ from 8 to 28 in the first stage of a LAS-ResNet brings very little improvement on V100. Based on the results in Figure 10, we can trade off between flexibility and efficiency by selecting appropriate $S$ for different models and hardware devices. For example, we can simply set $S_{\text{net}}$=8-4-7-1[5] in a LAS-ResNet-101 to achive realistic speedup. The accuracy-latency plots in Figure 7 also validate this observation. More results of our latency prediction model on the desktop-level GPU, GTX1080, are presented in Appendix C.1.

## 4.3 ImageNet classification results

We now empirically evaluate our proposed LASNet on the ImageNet dataset. The network performance is measured in terms of the trade-off between classification accuracy and inference efficiency. Both theoretical (*i.e.* FLOPs) and practical efficiency (*i.e.* latency) are tested in our experiments.

---

[5]We use this form to represnet the $S$ settings for the 4 stages of a network.

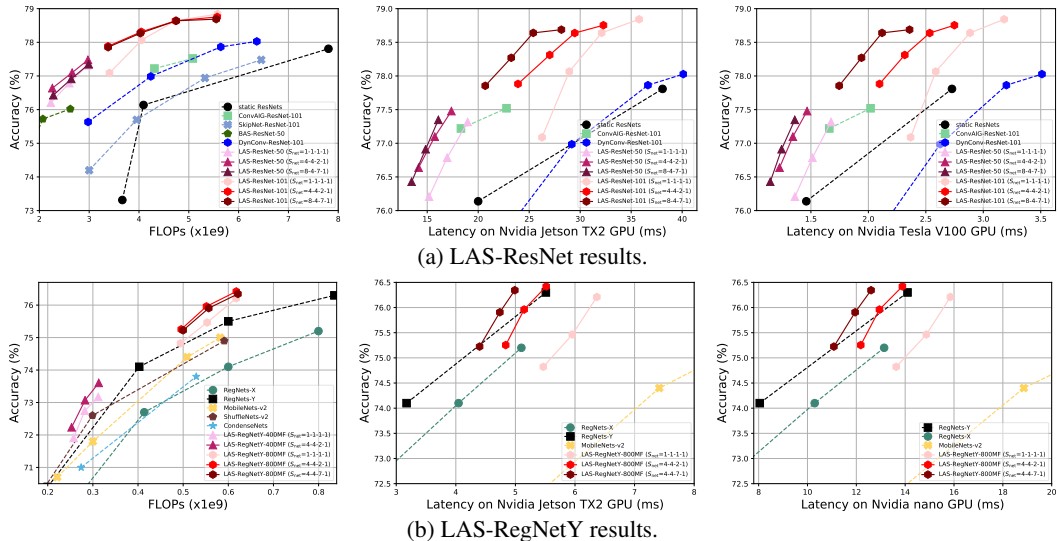

(a) LAS-ResNet results.

(b) LAS-RegNetY results.

Figure 7: Experimental results on ImageNet. The proposed coarse-grained spatially adaptive inference is tested on standard ResNets (a) and lightweight RegNets (b).

### 4.3.1 Standard baseline comparison: ResNets

We first establish our LASNet based on the standard ResNets [11]. Specifically, we build LAS-ResNet-50 and LAS-ResNet-101 by plugging our maskers in the two common ResNet structures.

**The baselines** include various types of dynamic inference approaches: 1) layer skipping (SkipNet [32] and Conv-AIG [30]); 2) channel skipping (BAS [1]); and 3) pixel-level spatial-wise dynamic network (DynConv [31]). For our LASNet, we compare various settings of the spatial granularity $S_{net}$. We set training targets (cf. Sec. 3.5) $t \in \{0, 4, 0.5, 0.6, 0.7\}$ for our dynamic models to evaluate their performance in different sparsity regimes. We apply the same operator fusion (Sec. 3.4) for both our models and the compared baselines [30, 31] for fair comparison.

**Results** are presented in Figure 7 (a). On the left we plot the relationship of accuracy *v.s.* FLOPs. It can be observed that our LAS-ResNets with different granularity settings significantly outperform the competing dynamic neural networks. Surprisingly, coarse-grained spatially adaptive inference ($S_{net}$=4-4-2-1 and $S_{net}$=8-4-7-1 for the 4 stages) can achieve even higher accuracy when consuming similar FLOPs on ResNets, despite the sacrificed flexibility compared to $S_{net}$=1-1-1-1. We conjecture that a larger $S$ is also beneficial to the optimization of maskers.

We compare the practical latency of three granularity settings in Figure 7 (a) predicted by our latency prediction model (middle on TX2 and right on V100). We can witness that although they achieve comparable theoretical efficiency (Figure 7 (a) left), larger $S$ is more hardware-friendly compared to the finest granularity. For example, the inference latency of LAS-ResNet-101 ($S_{net}$=1-1-1-1) is significantly higher than the ResNet-101 baseline on V100 (Figure 7 (a) right), even though its theoretical computation is much smaller than that of the static model. However, larger granularities ($S_{net}$=4-4-2-1 and $S_{net}$=8-4-7-1) can effectively improve the inference latency due to its lower burden on the memory access. Remarkably, the latency of ResNet-101 could be reduced by 36% and 46% on V100 and TX2 respectively without sacrificing the accuracy when $t$=0.4. The classification accuracy is increased by 1.9% with similar inference efficiency. It can be observed that the realistic speedup ratio $r_\ell$ is more close to the theoretical FLOPs ratio target $t$ on the less powerful TX2, because the latency is *computation-bounded* (*i.e.* the latency is mainly spent on computation) on such IoT devices. In contrast, there is a larger gap between practical and theoretical efficiency on the more powerful V100, as the latency is bounded by the memory access cost.

### 4.3.2 Lightweight baseline comparison: RegNets

We further evaluate our LASNet in lightweight CNN architectures, *i.e.* RegNets-Y [25]. Two different sized models are tested: RegNetY-400MF and RegNetY-800MF. Compared baselines include other types of efficient models, *e.g.*, MobileNets-v2 [28], ShuffletNets-v2 [23] and CondenseNets [16].

The results are presented in Figure 7 (b). The x-axis for the three sub-figures are the FLOPs, the latency on TX2, and the latency on the Nvidia nano GPU, respectively. We can observe that our

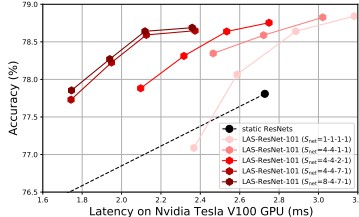

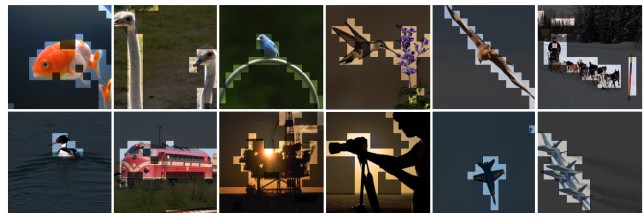

Figure 8: Ablation studies on $S$.

Figure 9: Visualization results.

method outperforms various types of static models in terms of the trade-off between accuracy and efficiency. More results on image classification are provided in Appendix C.2.

## 4.4 Ablation studies

We conduct ablation studies to validate the effectiveness of our *coarse-grained* spatially adaptive inference (Sec. 3.2) and operator fusion operations (Sec. 3.4).

**More granularities settings.** We test various granularity settings on LAS-ResNet-101 to examine the effects of $S$ in different stages. The results on the Tesla-V100 GPU are presented in Figure 8. It can be found that the finest granularity ($S_{net}$=1-1-1-1) leads to substantial inefficiency despite the reduced FLOPs (cf. Figure 7 (a) left). Coarse-grained spatially adaptive inference in the first two stages ($S_{net}$=4-4-1-1) effectively reduces the inference latency. We further increase $S$ in the third stage to 2 and 7, and this procedure consistently improves the realistic efficiency on the V100 GPU. This trend also holds on the GTX 1080 GPU (see the results in Appendix C.2.

**Operator fusion.** We investigate the effect of our operator fusion introduced in Sec. 3.4. One convolutional block in the first stage of a LAS-ResNet-101 ($S$=4, $r$=0.6) is tested. The results in Table 1 validate that every step of operator fusion benefits the practical latency of a block, as the overhead on memory access is effectively reduced. Especially, the fusion of the masker operation and the first convolution is crucial to reducing the latency.

Table 1: Ablation studies on operator fusion.

| Masker-Conv1x1 | Gather-Conv3x3 | Scatter-Add | Latency (μs) |
|:---:|:---:|:---:|:---:|
| ✗ | ✗ | ✗ | 163.2 |
| ✓ | ✗ | ✗ | 90.1 |
| ✓ | ✓ | ✗ | 86.7 |
| ✓ | ✓ | ✓ | **71.4** |

## 4.5 Visualization

We visualize the masks generated by our masker in the third block of a LAS-ResNet-101 ($S_{net}$=4-4-2-1) in Figure 12. The brilliant areas correspond to the locations of 1 elements in a mask, and the computation on the dimmed regions is skipped by our dynamic model. We can witness that the masker accurately locate the most task-related regions (even the tiny aircraft at the corner), which helps reduce the unnecessary computation on background areas. These resultsalso suggest that for the first stage, the granularity $S$=4 is sufficiently flexible to recognize the important regions, and a *win-win* can be achieved between accuracy and efficiency. Interestingly, the masker could select some objects that are *not labeled* for the sample, *e.g.*, the flower beside the hummingbird and the human holding the camera. This indicates that our spatial-wise dynamic networks can automatically recognize the regions with semantics, and their capability is not limited by the classification labels. This property is helpful in some downstream tasks, such as object detection (Sec. 4.6) and instance segmentation (Sec. 4.7), which require detecting multiple classes and objects in an image. More visualization results could be found in Appendix C.3.

## 4.6 COCO Object detection

We further evaluate our LASNet on the COCO [22] object detection task. The mean average precision (mAP), the average backbone FLOPs, and the average backbone latency on the validation set are used to measure the network performance. We test two commonly used detection frameworks: Faster R-CNN [27] with Feature Pyramid Network [20] and RetinaNet [21]. Thanks to the generality of our method, we can conveniently replace the backbones with ours pre-trained on ImageNet, and the whole models are finetuned on COCO with the standard setting for 12 epochs (see detailed

Table 2: Object detection results on the COCO dataset.

| Detection Framework | Backbone | Backbone FLOPs (G) | Backbone Latency (ms) | | | mAP (%) |
|---|---|---|---|---|---|---|
| | | | V100 | GTX1080 | TX2 | |
| Faster R-CNN | ResNet-101 (Baseline) | 141.2 | 39.5 | 119.3 | 729.4 | 39.4 |
| | LAS-ResNet-101 ($S_{net}$=4-4-2-1, $t$=0.6) | 90.7 | 33.8 | 90.6 | 524.9 | **40.3** |
| | LAS-ResNet-101 ($S_{net}$=4-4-2-1, $t$=0.5) | 79.3 | 30.7 | 82.5 | 477.1 | 39.8 |
| | LAS-ResNet-101 ($S_{net}$=4-4-7-1, $t$=0.5) | 79.5 | 29.0 | 79.9 | 464.8 | 40.0 |
| | LAS-ResNet-101 ($S_{net}$=4-4-7-1, $t$=0.4) | **67.9** | **25.3** | **69.1** | **401.3** | 39.5 |
| RetinaNet | ResNet-101 (Baseline) | 141.2 | 39.5 | 119.3 | 729.4 | 38.5 |
| | LAS-ResNet-101 ($S_{net}$=4-4-2-1, $t$=0.5) | 77.8 | 30.4 | 81.8 | 472.6 | **39.3** |
| | LAS-ResNet-101 ($S_{net}$=4-4-7-1, $t$=0.5) | 79.4 | 28.9 | 79.9 | 464.8 | **39.3** |
| | LAS-ResNet-101 ($S_{net}$=4-4-7-1, $t$=0.4) | **66.4** | **25.3** | **69.1** | **401.3** | 38.9 |

Table 3: Instance Segmentation results on the COCO dataset.

| Segmentation Framework | Backbone | Backbone FLOPs (G) | Backbone Latency (ms) | | | $AP^{mask}$ (%) | $AP^{box}$ (%) |
|---|---|---|---|---|---|---|---|
| | | | V100 | GTX1080 | TX2 | | |
| Mask R-CNN | ResNet-101 (Baseline) | 141.2 | 39.5 | 119.3 | 729.4 | 36.1 | 40.0 |
| | LAS-ResNet-101 ($S_{net}$=4-4-2-1, $t$=0.5) | 80.5 | 31.1 | 83.3 | 481.9 | **37.0** | **41.0** |
| | LAS-ResNet-101 ($S_{net}$=4-4-2-1, $t$=0.4) | 69.2 | 27.9 | 74.8 | 431.6 | 36.1 | 40.0 |
| | LAS-ResNet-101 ($S_{net}$=4-4-7-1, $t$=0.4) | **68.8** | **25.8** | **70.9** | **411.8** | 36.2 | 40.0 |

setup in Appendix B.3). The input images are resized to a short side of 800 and a long side not exceeding 1333. The results of our LAS-ResNet-101 with different $S_{net}$ settings are presented in Table 2. We can observe from the results that when setting the training target as 0.4, the latency of our LAS-ResNet-101 with $S_{net}$=4-4-7-1 is significantly lower than the static baseline on all devices without sacrificing mAP in both detection frameworks. With a larger training targets, our LASNet can increase the mAP by 0.9% and 0.8% in Faster R-CNN and RetinaNet respectively, while still being faster than the baseline method.

## 4.7 COCO instance segmentation

We also present the results of instance segmentation on COCO, which demonstrate the effectiveness of our LASNet on the dense prediction task. From the results in Table 3, we can observe that when setting the training target as 0.4, the Mask R-CNN [10] models ($S_{net}$=4-4-2-1 and $S_{net}$=4-4-7-1) runs faster on all tested hardware devices without sacrificing the performance. With a training target of 0.5, the $AP^{mask}$ and $AP^{box}$ of the Mask R-CNN model could be increased by 0.9% and 1.0% respectively while still running faster than the baseline method.

## 5 Conclusion

In this paper, we propose to build *latency-aware* spatial-wise dynamic networks (LASNet) under the guidance of a *latency prediction model*. By simultaneously considering the algorithm, the scheduling strategy and the hardware properties, we can efficiently estimate the practical latency of spatial-wise dynamic operators on arbitrary computing platforms. Based on the empirical analysis on the relationship between the latency and the *granularity* of spatially adaptive inference, we optimize both the algorithm and the scheduling strategies to achieve realistic speedup on many multi-core processors, *e.g.*, the Tesla V100 GPU and the Jetson TX2 GPU. Experiments on image classification, object detection and instance segmentation tasks validate that the proposed method significantly improves the practical efficiency of deep CNNs, and outperforms various competing approaches.

## Acknowledgement

This work is supported in part by the National Key R&D Program of China under Grant 2020AAA0105200, the National Natural Science Foundation of China under Grants 62022048 and the Tsinghua University-China Mobile Communications Group Co.,Ltd. Joint Institute..

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
