# Appendix

## A  Latency prediction model.

As the dynamic operators in our method have not been supported by current deep learning libraries, we propose a latency prediction model to efficiently estimate the real latency of these operators on hardware device. The inputs of the latency prediction model include: 1) the structural configuration of a convolutional block, 2) its activation rate $r$ which decides the computation amount, 3) the spatial granularity $S$, and 4) the hardware properties mentioned in Table 4. The latency of a dynamic block is predicted as follows.

**Input/output shape definition**. The first step of predicting the latency of an operation is to calculate the shape of input and output. Taking the gather-conv2 operation as an example, the input of this operation is the activation with the shape of $C_{in} \times H \times W$, where $C_{in}$ is the number of input channels, and $H$ and $W$ are the resolution of the feature map. The shape of the output tensor is $P \times C_{out} \times S \times S$, where $P$ is the number of output patches, $C_{out}$ is the number of output channels and $S$ is the spatial granularity. Note that $P$ is obtained based on the output of our maskers.

**Operation-to-hardware mapping.** Next, we map the operations to hardware. As is mentioned in the paper, we model a hardware device as multiple processing engines (PEs). We assign the computation of each element in the output feature map to a PE. Specifically, we consecutively split the output feature map into multiple *tiles*. The shape of each tile is $T_P \times T_C \times T_{S1} \times T_{S2}$. These split tiles are assigned to multiple PEs. The computation of the elements in each tile is executed in a PE. We can configure different shapes of tiles. In order to determine the optimal shape of the tile, we make a search space of different tile shapes. The tile shape has 4 dimensions. The candidates of each dimension are power-of-2 and do not exceed the corresponding dimension of the feature map.

**Latency estimation.** Then, we evaluate the latency of each tile shape in the search space and select the optimal tile shape with the lowest latency. The latency includes the *data movement* latency and the *computation* latency: $\ell = \ell_{\mathrm{data}} + \ell_{\mathrm{computation}}$.

1) *Data movement latency* $\ell_{\mathrm{data}}$. The estimation of the latency for data movement requires us to model the memory system of a hardware device. We model the memory system of hardware as a three-level architecture [12]: off-chip memory, on-chip global memory, and local memory in PE. The input data and weight data are first transferred from the off-chip memory to the on-chip global memory. We assume the hardware can make full use of the off-chip memory bandwidth to simplify the latency prediction model.

After that, the data used to compute the output tiles is transferred from on-chip global memory to the local memory of each PE. The latency of data movement to local memory is estimated by its *bandwidth* and *efficiency*. We assume each PE only moves the corresponding input feature maps and weights once to compute a output tile so as to simplify the prediction model. The input data movement latency $\ell_{\mathrm{in}}$ is calculated by adding the time from off-chip memory to on-chip global memory and the time from on-chip global memory to local-memory together: $\ell_{\mathrm{in}} = \ell_{\mathrm{off2on}} + \ell_{\mathrm{global2local}}$. Contrary to the input data, the output data $\ell_{\mathrm{out}}$ are moved from local memory to on-chip global memory and then to off-chip memory: $\ell_{\mathrm{out}} = \ell_{\mathrm{local2global}} + \ell_{\mathrm{on2off}}$. We calculate the total data movement latency by adding the input and output data movement latency together: $\ell_{\mathrm{data}} = \ell_{\mathrm{in}} + \ell_{\mathrm{out}}$.

The latency of data movement is affected by the granularity $S$: when the granularity $S$ is small, the same input data has a higher probability of being sent to multiple PEs to compute different output patches, which significantly increases the number of on-chip memory movement. And due to the small amount of data transmitted each time and the data is randomly distributed, the efficiency of data movement will be low. This accounts for our experiment results in the paper that *a larger $S$ will effectively improve the practical efficiency*.

2) *Computation latency* $\ell_{\mathrm{computation}}$. The computation latency of each tile is estimated using the PE's *maximum throughput of FP32 computation* and the *FLOPs* of computing an output tile. The total computation latency can be obtained according to the number of tiles and the number of PEs.

To summarize, our latency prediction model can predict the real latency of dynamic operators by considering both the *data movement* cost and the *computation* cost. Guided by the latency prediction model, we propose our LASNets with coarse-grained spatially adaptive inference ($S > 1$). It is validated in our paper that LASNets achieve better efficiency than previous approaches [35, 31]

$(S = 1)$, as it effectively reduces the data movement latency, which is rarely considered by other researchers.

## B    Detailed experimental settings

In this section, we present the detailed experiment settings which are not provided in the main paper due to the page limit.

### B.1    Latency prediction

**Hardware properties** considered by our latency prediction model include the number of processing engines (#PE), the floating-point computation in a processing engine (#FP32), the frequency and the bandwidth. We test four types of hardware devices, and their properties are listed in Table 4.

Table 4: Hardware properties.

| Name | #PE | #FP32 | frequency (MHz) | bandwidth (G) |
|---|---|---|---|---|
| Nvidia Tesla V100 | 80 | 64 | 1500 | 700 |
| Nvidia GTX1080 | 20 | 64 | 1700 | 320 |
| Nvidia Jetson TX2 | 2 | 128 | 1300 | 59.7 |
| Nvidia Nano | 1 | 128 | 921 | 25.6 |

It could be found that the server GPU V100 is the most powerful hardware device, especially with the most number of processing engines (#PE). Therefore, spatially adaptive inference could easily fall into a *memory-bounded* operation on V100 due to its high parallelism. Our experiment results in Figure 7 (a) and Figure 8 in the paper can reflect this phenomenon: the more flexibility the computation is, the harder to improve the practical efficiency.

In contrast, on the less powerful computing devices such as the IoT devices, the real acceleration is close to the theoretical effect (compare Figure 7 (a) left with Figure 7 (a) middle).

**Operator fusion.**

1) *Fusing the masker and the first convolution.* We mentioned in Sec. 3.4 of the paper that the masker operation is fused with the first 1×1 convolution in a block to reduce the cost on memory access. This is feasible because the two operators share the same input feature, and their convolutional kernel sizes are both 1×1.

Note that during the inference stage, we only need to perform $\arg\max$ along the channel dimension of a mask $\mathbf{M} \in \mathbb{R}^{2 \times H \times W}$ to obtain the positions of the gathered pixels. Therefore, we can reduce the output channel number of our maskers from 2 to 1 since the convolution is a linear operation:

$$[\mathbf{x} * \mathbf{W}]_{:,:,0} > [\mathbf{x} * \mathbf{W}]_{:,:,1} \iff \mathbf{x} * (\mathbf{W}_{:,:,0} - \mathbf{W}_{:,:,1}) > 0. \tag{3}$$

Afterwards, we fuse the masker with the first convolution layer by performing once convolution whose output channel number is $C + 1$, where $C$ is the original output width of the first convolution. The output of this step is split into a feature map (for further computation) and a mask (for obtaining the index for gathering). Such operator fusion avoids the repeated reading the input feature, and helps reduce the inference latency (see Table 1 in the paper).

2) *Fusing the gather operation and the dynamic convolution.* To facilitate the scheduling on hardware devices with multiple PEs, the masker generates the indices of activated patches instead of sparse mask at inference time. In this way, it is easy to evenly distribute the computation of output patches to different PEs, thus avoiding unbalanced computation of PEs. Each element in the indices represents the index of an activated patch. PE fetches the input data from the corresponding positions on the feature map according to the index. The output patches could be densely stored in memory. Such operator fusion benefits the contiguous memory access and parallel computation on multiple PEs.

3) *Fusing the scatter operation and the add operation.*

Similar to the previous operation, each PE fetches a tile of data from the residual feature map according to the index, adds them with the corresponding feature map from previous dynamic

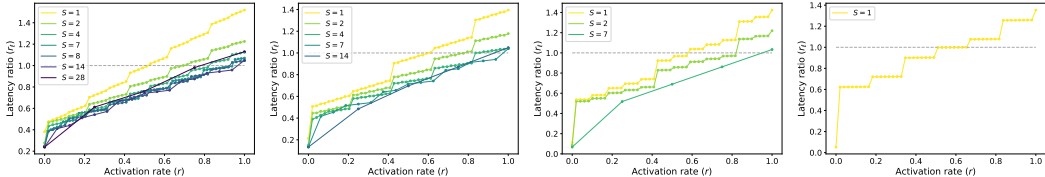

(a) Relationship between $r_\ell$ and $r$ for LAS-ResNet blocks on Nvidia GeForce GTX1080.

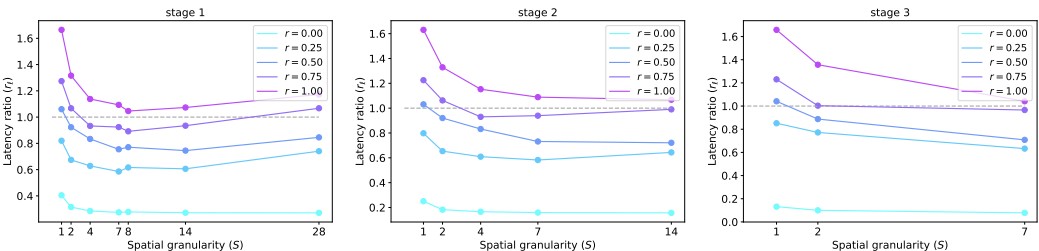

(b) Relationship between $r_\ell$ and $S$ for LAS-ResNet blocks on Nvidia GeForce GTX1080.

Figure 10: Latency prediction results of LAS-ResNet blocks on the Nvidia GTX1080 GPU.

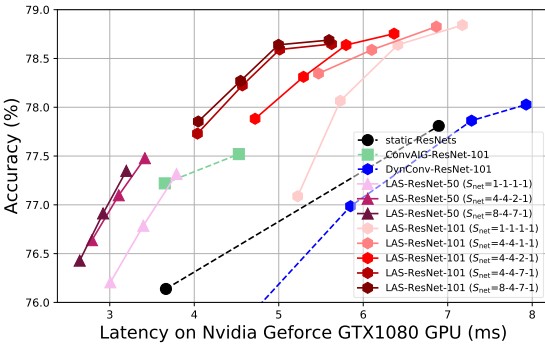

Figure 11: Experimental results on the ImageNet classification task.

convolution, and then stores the results to the corresponding position on the residual feature map according to the index. This optimization can significantly reduce the costs on memory access.

**Speed test.** We test the latency on real hardware devices to evaluate the accuracy of our latency prediction model. On GPUs, we use TVM [2] and CUDA (version 11.6) for code generation and compilation respectively. The results in Fig. 4 of the paper validate the effectiveness of our model.

## B.2    ImageNet classification

As mentioned in the paper, we use pre-trained CNN models in the official torchvision website to initialize our backbone parameters, and finetune the overall models for 100 epochs. The initial learning rate is set as $0.01 \times$batch size/128, and decays with a cosine shape. The training batch size is determined on the model size and the GPU memory. For example, we train our LAS-ResNet-101 on 8 RTX 3090 GPUs with the batch size of 512, and the batch size for LAS-ResNet-50 is doubled. We use the same weight decay and the standard data augmentation as in the RegNet paper [25]. For our own hyper-parameter $\tau$ in Eq. (1) of the paper, this Gumbel temperature $\tau$ exponentially decreases from 5 to 0.1 in the training procedure. For the training hyper-parameter in Eq. (2), we simply fix $\alpha = 10, \beta = 0.5$ and $T = 4.0$ for all dynamic models. We conduct a very simple grid search with a RegNet for $\beta \in \{0.3, 0.5\}$ and $T \in \{1.0, 4.0\}$ to determine their values.

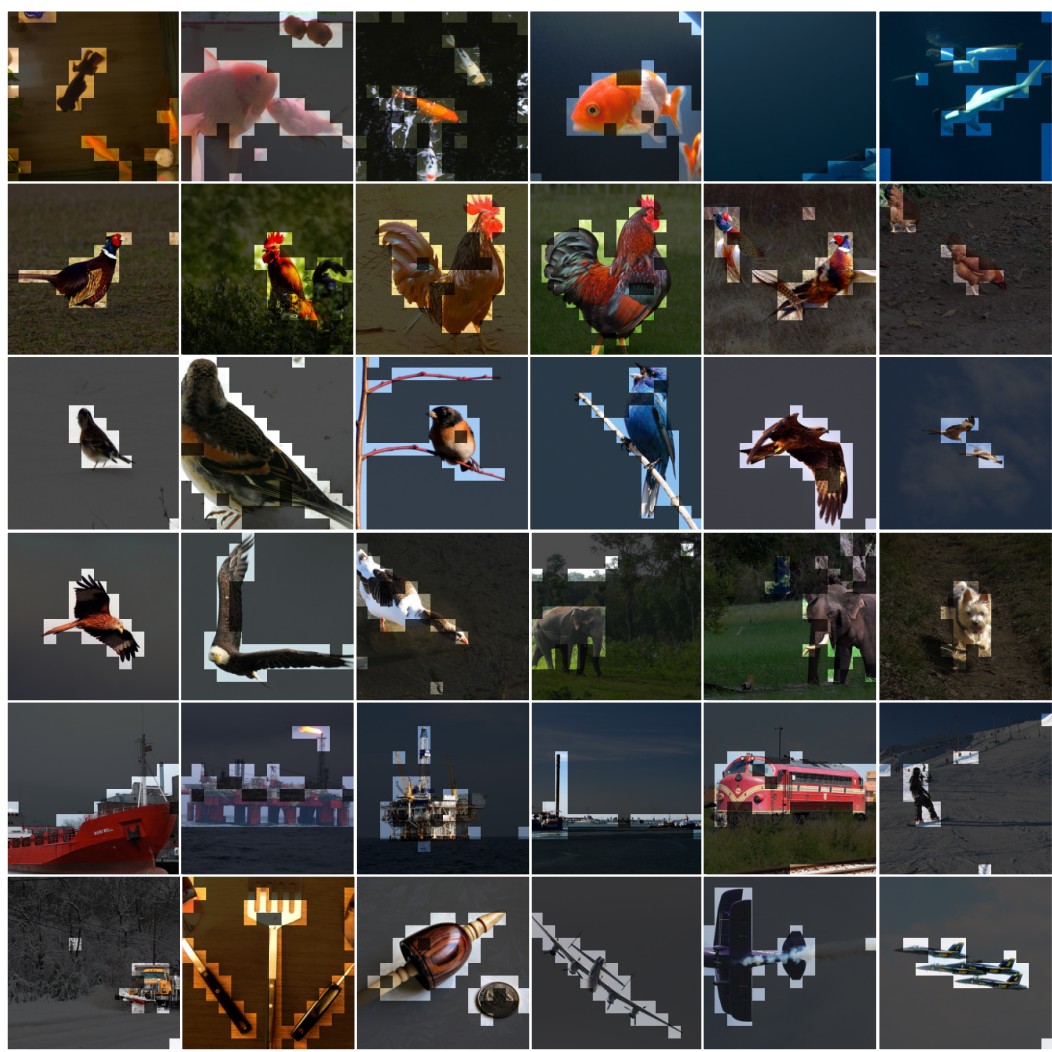

Figure 12: Visualization results.

### B.3 COCO object detection & instance segmentation

We use the standard setting suggested in [20, 21, 10], except that we decrease the learning rate for our pre-trained backbone network. We simply set a learning rate multiplier 0.5 for Faster R-CNN [27], 0.2 for RetinaNet [21] and 0.5 for Mask R-CNN [10]. As for the additional loss items, the hyper-parameters are kept the same as training our classification models, except that the temperature is fixed as 0.1 in the 12 training epochs.

## C More experimental results

In this section, we report more experimental results which are not presented in the main paper.

### C.1 Latency prediction

In Figure 5 and Figure 6 of the paper, we report the latency prediction results of LAS-ResNet on V100 and TX2. Here we present the results on GTX1080 (Figure 10). It can be found that $S_{net}$=8-4-7-1 (which is the same to the optimal setting on V100 in the paper) will lead to faster inference on GTX1080. This is reasonable since GTX1080 has a large #PE than those IoT devices, and requires more contiguous memory access (a larger $S$ for coarse-grained spatially adaptive inference) to achieve realistic speedup ($r_\ell < 1$). The accuracy-latency curves in Figure 11 further validate this observation.

## C.2 ImageNet classification

**Results on GTX1080 GPU.** In Figure 7 of the paper, we report the ImageNet classification results of LAS-ResNets on V100 and TX2. Here we present the results of LAS-ResNet on GTX1080 (Figure 11). From the results in Figure 11, we can get the same conclusion as in the paper, that a proper large $S$ is more parallel-friendly than the finest granularity. Remarkably, the latency of ResNet-101 could be reduced by 41% without sacrificing the accuracy on GTX1080 under the spatial granularity setting of $S_{\text{net}}$=8-4-7-1. With a similar inference latency, the accuracy of a static ResNet-101 could be increased by 1.0% by our LAS-ResNet-101 ($S_{\text{net}}$=4-4-2-1).

**Results of the extreme situation of** $S_{\text{net}}$=56-28-14-7. We mentioned in the paper that when we set $S_{\text{net}}$=56-28-14-7, the spatially adaptive inference paradigm will reduce into layer skipping. We experiment on ResNet-50, and find that although being slightly faster (13ms) than our LAS-ResNet-50 ($S_{\text{net}}$=4-4-2-1, 16ms), the accuracy of LAS-ResNet-50 with $S_{\text{net}}$=56-28-14-7 could be significantly degraded from 76.6% (ours, $S_{\text{net}}$=4-4-2-1) to 76.1%. Therefore, we mainly focus on the discussion of spatially adaptive inference in this paper.

## C.3 More visualization results

In addition to Figure 9 in the paper, here we present more visualization results of the regions selected by our masker in the 3-rd block of a LAS-ResNet-101 ($S_{\text{net}}$=4-4-2-1) in Figure 12, which demonstrate that our spatially adaptive inference paradigm can effectively locate the most task-related areas in image features, and reduce the unnecessary computation on those background areas.

# D Limitations

The current limitations of our work include the following aspects:

1) the latency-ware co-designing framework is only constructed for spatial-wise dynamic networks. Support for more types of dynamic models (*e.g.* channel skipping) will be explored in the future;

2) To achieve faster inference, the spatial masks are defined the same for all input/output channels, which might limit the flexibility of adaptive inference. Future work may explore more flexible forms of dynamic computation;

3) The combination with other acceleration techniques such as Winograd, and the implementation on more CNN backbones may be worth studying in the future.

**Social impact.** Our work can help reduce the inference cost of deep CNNs, but the training of our models might potentially increase the carbon emissions.