# OpenReview forum: "Latency-aware Spatial-wise Dynamic Networks"
_NeurIPS.cc/2022/Conference — NeurIPS 2022 Accept_

### Official Review · Reviewer_MLaG · 2022-07-10

**Rating:** 5
**Confidence:** 3
**Soundness:** 3 good
**Presentation:** 2 fair
**Contribution:** 2 fair

**Summary:**

This submission introduces a spatially dynamic neural network approach, developed in a latency-aware manner that can generate realistic speed-ups during inference. The proposed methodology allows to dynamically skip computation at different spatial regions of the feature maps at a coarser granularity (of blocks of pixels), directly generalising the widely studied pixel-level approaches from the literature. This leads to more regular computation pattern and reduces the overheads, to maximise the attainable inference speed gains. Additionally, the granularity of the pixel blocks is tuned in a latency-aware manner to balance the speed-accuracy trade-off. Results indicate that the proposed approach is effectively improving efficiency, across hardware platforms and models for image classification and object detection tasks.

**Questions:**

1. How is the latency predictor implemented? Is it trainable, or based on an analytical performance model/ cycle accurate simulator? Can it capture the effect of more complex operations, such as dilated or separable convs, skip connections etc?
2. The case where a large S can emulate layer skipping should be considered in the ablation.
3. How would the proposed approach be applied to dense tasks, such as semantic segmentation, dense prediction etc ?
4. Is there any benefit on allowing overlapping pixel blocks to be considered during masking? Qualitative results indicate that such a setting may allow for larger values of S.

**Limitations:**

Some limitations are discussed in the conclusion section. The authors are encouraged to enhance this discussion.

**Strengths And Weaknesses:**

Strengths:
- The paper addresses a very important real-world issue, which is well-motivated and demonstrated experimentally.
- The proposed approach, although rather simple, is able to generate realistic speed-ups with no effect in accuracy, showcasing improved efficiency compared to baselines.
- Useful insights are provided in the experiments analysis.

Weaknesses (see questions for more details):
- The novelty of the proposed approach is limited, as it directly generalises a widely studied problem, going from pixel-level to block-level. Similar generalisation in the context of early-exit networks has recently been studied in the literature:
  - Liu, Zhuang, et al. "Anytime Dense Prediction with Confidence Adaptivity." International Conference on Learning Representations. 2022.
-  The description of the latency prediction model (Sec 3.3) is very high-level, making it hard to justify its contribution .
- Some interesting aspects of this work (eg the generic formulation combining spatial computation skipping and layer skipping - Sec 3.2) are only theoretically mentioned, but not experimentally studied.
- Some of the claimed contributions and ablation studies are mostly comprise technical/implementation aspects (Sec 3.4).
- Evaluation is focused on Classification/detection, but it is unclear how the proposed methodology can be applied to more dense CV tasks, such as instance/semantic segmentation. The reviewer believes that this mapping won't be intuitive and this should be discussed in the limitations of the proposed approach.

Presentation
- Syntax/grammatical errors exist sporadically across the manuscript, mostly in Sec1, 3.3 and 4.5
- Some aspects of the proposed methodology are described in rather high-level, making it impossible to reproduce.

---

> ### Author Response · Authors · 2022-08-02
> **Response to Reviewer MLaG (part3)**
>
> ### Questions
>
> > How is the latency predictor implemented? Is it trainable, or based on an analytical performance model/ cycle accurate simulator? Can it capture the effect of more complex operations, such as dilated or separable convs, skip connections etc?
>
> - The latency predictor is an analytical performance model and it is not trainable. We model the behavior of the accelerators and we define each operation using nested for-loop. We map the operation to hardware using various schedules and evaluate the latencies and choose the optimal schedule. The model considered the number of PE, the off-chip memeory bandwidth, the on-chip memeory bandwidth, and the throughput of FP32 computation of a PE to calculate the data transfer time and the computation time. **We omitted these details in the main paper for simplicity, and have updated the paper and the supplementary material to include a more detailed description of our latency prediction model.**
> - We now only implement the operations required by our experiments. It supports the group convolution in RegNets (and separable convolutions, of course). Skip connection is supported. Dilated convs will be included in the future.
>
>
>
> > The case where a large S can emulate layer skipping should be considered in the ablation.
>
> - Please see our response to Weakness 3.
>
>
>
> > How would the proposed approach be applied to dense tasks, such as semantic segmentation, dense prediction etc ?
>
> - Please see our response to Weakness 5.
>
>
>
> > Is there any benefit on allowing overlapping pixel blocks to be considered during masking? Qualitative results indicate that such a setting may allow for larger values of S.
>
> - An interesting question.  As far as we know, this paradigm has not been studied by researchers in this field. To our understanding, the proposed formulation will bring more flexibility in spatially adaptive inference, but may also have some implementation issues, since the computation of a pixel will be determined by multiple elements in a mask. This might raise considerable challenges in searching for the optimal scheduling during inference. Moreover, the difficulty of training the dynamic model will be increased, as the current Gumbel Softmax reparameterization technique might be impractical in this situation. Moreover, as our results in Fig 7 show, it's not always beneficial to increase S, especially on less powerful devices such as TX2 and Nvidia Nano. In all, we believe this is a topic worth exploring in the future.
>
>
>
> ### Limitations
>
> > Some limitations are discussed in the conclusion section. The authors are encouraged to enhance this discussion.
>
> -  A more detailed analysis is updated in the supplementary material.

---

> > ### Comment · Reviewer_MLaG · 2022-08-05
> > **Discussion**
> >
> > Thank you for the thorough rebuttal and useful insights provided.
> >
> > Given the updated description you provided on the latency predictor, can you elaborate more its benefits to other estimators, mostly examined in the NAS community, e.g.:
> >
> > [1] Cai, Han, Ligeng Zhu, and Song Han. "Proxylessnas: Direct neural architecture search on target task and hardware.", ICLR 2019
> >
> > [2] Dai, Xiaoliang, et al. "Chamnet: Towards efficient network design through platform-aware model adaptation.", CVPR 2019
> >
> > [3] Li, Chaojian, et al. "Hw-nas-bench: Hardware-aware neural architecture search benchmark.", ICLR 2021
> >
> > [4] Wu, Bichen, et al. "Fbnet: Hardware-aware efficient convnet design via differentiable neural architecture search.", CVPR 2019

---

> > > ### Author Response · Authors · 2022-08-06
> > > **Discussion on the latency prediction model**
> > >
> > > Thanks for replying!
> > >
> > > - First, we would like to clarify that the most significant difference between our predictor and the mentioned estimators is that:
> > >   - The mentioned estimators are built based on the latency data **tested** on targeted hardware devices. The estimators are implemented either as **a lookup table or a trainable neural network**. These approaches are practical only for **static** networks, as all the static operations are well supported by existing libraries (e.g., cuDNN). We assume it is more appropriate to call it "**testing and querying/approximating** the latency" by treating the hardware as a **black box** in the NAS community.
> > >   - In contrast, we predict the latency by treating hardware as a **white box** and **modeling its detailed behaviors and properties**.
> > >     - **Why**? Currently, no software library provides the efficient implementation of our **dynamic** operators. An efficient implementation of the dynamic operators requires laborious code optimization for each granularity (S), each activation rate (r), and each hardware device. It would be quite **time-consuming and engineering** if we deploy the dynamic operators on specific devices and test the latency as in the NAS community.
> > >     - **How**? We are able to accurately predict the latency because we comprehensively consider the cost of both **data movement** and **parallel computation** in the latency prediction procedure. We model a hardware device with a three-level architecture (Fig. 3 of the paper) and estimate the latency based on detailed hardware parameters, such as the frequency, the number of processing engines, the bandwidth, etc., as mentioned in the supplementary material.
> > > - The advantages of our predictor include the following aspects:
> > >   - **Dynamic operators are supported for the first time**. As mentioned before, the current libraries do not support dynamic operators yet. Our latency prediction model enables the analysis of dynamic operators for the first time;
> > >   - **Efficiency**. We can conveniently compare the performance of different granularity settings on arbitrary hardware devices **without a laborious deployment** (especially for dynamic operators). Furthermore, we only need to adjust the hardware parameters to obtain the performance of an operator on different hardware devices;
> > >   - **Our predictor can play as an analytic tool**. We can easily analyze the key factor to the latency of each operator **(e.g., memory bounded or computation bounded)**. However, the mentioned estimators in the NAS community can only provide the overall latency result and tell us which static operator is faster. Our predictor can **guide** the efficient implementation (e.g., the operator fusion) of dynamic operators.  For example, when the memory access (data movement) contributes more to the overall latency of the masker and conv1, we can merge them to reduce the memory access cost. In this way, the overall latency can be reduced even if the computation is increased.
> > > - Of course, our latency prediction model **can also be easily used in NAS**.
> > > - Moreover, inspired by your comment, we believe that our proposed predictor can facilitate the automatic searching of dynamic networks (**NAS for dynamic networks**). For example, we could include dynamic operators in search space. This could further expand our work since we only focus on the granularity of spatially adaptive computation in this paper. More types of dynamic operators (e.g., channel skipping) will be supported in the future.

---

> > > ### Author Response · Authors · 2022-08-08
> > > **Another reminder of discussion**
> > >
> > > Thanks again for your time!
> > >
> > > We kindly ask the reviewer to reassess the paper in light of this comment if it clears things up. We are happy to answer more if you have any remaining concerns or questions.

---

> ### Author Response · Authors · 2022-08-02
> **Response to Reviewer MLaG (part2)**
>
>
>
> > Some of the claimed contributions and ablation studies are mostly comprise technical/implementation aspects (Sec 3.4).
>
> - As mentioned before, we tackle an unexplored (yet very important) problem in dynamic neural networks. **The main contribution is not just designing a new network architecture, but proposing our latency-ware co-designing framework.** This has also been appreciated by the other two reviewers. Moreover, we have proposed **a new analytic method (latency prediction model) to efficiently and accurately evaluate the real latency of dynamic operators**. This model **guides** our implementation improvements (Sec. 3.4) and experiment design (Sec. 4.2). We believe that the overall framework can not only provide insights for the research community, but also facilitate the deployment of dynamic deep networks in practice.
>
>
>
> > Evaluation is focused on Classification/detection, but it is unclear how the proposed methodology can be applied to more dense CV tasks, such as instance/semantic segmentation. The reviewer believes that this mapping won't be intuitive and this should be discussed in the limitations of the proposed approach.
>
> - First, it has been proved by many previous works (e.g. reference [7,27,31] in the paper) that spatial-wise dynamic networks can be easily applied to dense prediction tasks. Our method can also work on segmentation tasks, as it is very convenient to substitute the backbone network in popular frameworks (such as mask R-CNN) with our LASNets. Since our main contribution is not designing a new powerful backbone, we did not report the experiment results on dense prediction tasks in the original submission.
> - We have tested our method in the Mask R-CNN framework on COCO instance segmentation. The results are listed in the table below, and are included in the supplementary material (Table 3). It can be observed that our LAS-ResNet101 significantly outperforms the baseline model. For example, to achieve comparable performance in AP^mask and AP^box, our LAS-ResNet101 (S=4-4-2-1, t=0.4) runs **~51% faster** than the static ResNet101 on TX2. Moreover, LAS-ResNet101 (S=4-4-2-1, t=0.5) can improve the AP^box and AP^mask of baseline by 1.0% (40.0% to **41.0%**) and 0.9% (36.1% to **37.0%**) while still running faster on GPUs.
>
> | Segmentation Framework | Backbone                               | Backbone FLOPs (G) | Backbone Latency (ms) on V100 | Backbone Latency (ms) on GTX1080 | Backbone Latency (ms) on TX2 | AP^mask (%) | AP^box (%) |
> | :--------------------: | -------------------------------------- | ------------------ | ----------------------------- | -------------------------------- | ---------------------------- | ----------- | ---------- |
> |                        | ResNet-101 (Baseline)                  | 141.2              | 39.2                          | 118.0                            | 720.7                        | 36.1        | 40.0       |
> |                        | LAS-ResNet-101 (S_net =4-4-2-1, t=0.4) | 69.2               | 42.4                          | 81.2                             | **352.6**                    | 36.1        | 40.0       |
> |       Mask R-CNN       | LAS-ResNet-101 (S_net =4-4-2-1, t=0.5) | 80.5               | 48.5                          | 93.3                             | 404.7                        | **37.0**    | **41.0**   |
> |                        | LAS-ResNet-101 (S_net =4-4-7-1, t=0.4) | **68.8**           | **33.1**                      | **76.1**                         | 391.6                        | 36.2        | 40.0       |
> |                        | LAS-ResNet-101 (S_net =4-4-7-1, t=0.5) | 82.0               | 38.0                          | 88.3                             | 457.4                        | 36.6        | 40.3       |
>
>
>
> > Presentation issues
>
>
> - We have revised the paper and the supplementary material to include more details. The language will be further polished.

---

> ### Author Response · Authors · 2022-08-02
> **Response to Reviewer MLaG (part1)**
>
> Thanks for your contrustive comments. The mentioned weaknesses and questions are addressed as follows:
>
> ### Weaknesses
>
> > The novelty of the proposed approach is limited, as it directly generalises a widely studied problem, going from pixel-level to block-level. Similar generalisation in the context of early-exit networks has recently been studied in the literature:
> >
> > - Liu, Zhuang, et al. "Anytime Dense Prediction with Confidence Adaptivity." International Conference on Learning Representations. 2022.
>
> - Actually, our work significantly differs from the mentioned reference (and many previous works) in the following aspects:
>   - **The studied problems and the motivations are totally different:** the mentioned reference proposes a **hardware-agnostic** algorithm for semantic segmentation. In contrast, we explore a problem which is rarely studied by previous researchers: improving the **practical** efficiency of dynamic backbone networks by **simultaneously considering the algorithm, the scheduling strategy and the hardware properties**.
>   - **The research approaches are different:** the mentioned work performs early exiting in segmentation networks, while our LASNet is more flexible and general which could be implemented in arbitrary backbone architectures.
>     - Moreover, we propose a **latency prediction model** to efficiently estimate the real latency of dynamic operators, and the algorithm design is guided by this latency predictor rather than the theoretical computation as done in previous works. To the best of our knowledge, we are the first to propose a latency prediction model which supports dynamic operators.
>   - **The products of our works are different:** like many previous works, the mentioned paper proposes an algorithm/network architecture. However, our main contribution is the **latency-aware co-designing framework**, which could be widely used by other researchers for designing various types of static/dynamic networks in the future.
>
>
>
> > The description of the latency prediction model (Sec 3.3) is very high-level, making it hard to justify its contribution .
>
> - Because the latency prediction model considers many hardware details (please also refer to our response to Question 1), we did not describe these details in the paper for simplicity. We have **updated the paper (Sec 3.3) and the supplementary material to describe more details of the latency prediction model**, and the code will be released upon acceptance.
> - To summarize, **the latency prediction model estimates the latency of a dynamic operator by considering both the data movement latency and the computation latency.**  This latency prediction model enables us to **accurately estimate the real latency of dynamic operators, which has not been explored by other researchers as far as we know**. The predictor plays an important role in our latency-aware co-design framework, as **it guides both our algorithm design and the scheduling optimization**.
>
>
>
> > Some interesting aspects of this work (eg the generic formulation combining spatial computation skipping and layer skipping - Sec 3.2) are only theoretically mentioned, but not experimentally studied.
>
> - We omitted the detailed discussion for S=56-28-14-7 (the coarsest granularity) in the paper because spatially adaptive inference will reduce into a layer skipping operation in this situation. However, the main purpose of this paper is to explore the practical efficiency of the former paradigm.
> - We empirically compare our models with the mentioned variant (with similar computational cost). It has been observed that although being slightly faster (13ms) than our LAS-ResNet-50 (S=4-4-2-1, 16ms) the accuracy could be significantly degraded from **76.6%** to **76.1%**.
> - Moreover, we have witnessed that when consuming similar computational cost and achieving similar accuracy (~78.8%) on ImageNet, our LAS-ResNet101 with S=4-2-2-1 (the proposed coarse-grained spatially adaptive computation, t=0.5 AP^box: **41.0%**, AP^mask: **37.0%**) significantly outperforms the variant with S=56-28-14-7 (layer skipping, t=0.5, AP^box: **40.4%**, AP^mask: **36.5%**) on COCO instance segmentation. When achieving comparable performance, the layer skipping scheme (S=56-28-14-7, t=0.6 AP^box=40.9%, AP^mask=36.8%) will run slower than our model (S=4-2-2-1, t=0.5, AP^box=**41.0%**, AP^mask=**37.0%**) by ~24.4% on TX2 (**535.6 ms** vs **404.7 ms**).
> - Based on the above experiments, we conjecture that **the extreme situation of dropping an entire feature map may be too aggressive, which will degrade the network performance.** In contrast, our spatially dynamic computation extracts better dense features, which is beneficial to both classification and downstream tasks. Therefore, we mainly focus on exploring the granularity settings in the spatial-wise dynamic computation paradigm.

---

> ### Author Response · Authors · 2022-08-09
> **Have our responses addressed the reviewer's concerns?**
>
> Dear Reviewer,
>
> We wanted to reach out to see if our most recent reply and paper revisions have addressed the concerns. With only ~17 hours left in the rebuttal period, we were hoping that the reviewer could confirm that the revisions have addressed the concerns about the problem statement, further clarify the concerns, and/or provide actionable suggestions for further improving the paper. We would be happy to further revise the paper to clarify any concerns or continue the discussion regarding any remaining concerns.
>
> We thank the reviewer for all their help so far in improving the paper!

---

### Official Review · Reviewer_BmLK · 2022-07-10

**Rating:** 6
**Confidence:** 4
**Soundness:** 3 good
**Presentation:** 3 good
**Contribution:** 3 good

**Summary:**

This paper presents a dynamic inference paradigm based on selective inference of convolutions on the spatial dimension. Specifically, for each convolutional block, it uses a masker layer to predict the masked region of the feature map and only conduct convolution on the masked region. Such convolution has less computation than a full convolution and improves efficiency of the model. Moreover, it utilizes a latency prediction model to estimate the latency of different mask operations. With a differentiable training framework, the model can be optimized according to configurations different hardware platform.

**Questions:**

See above.

**Limitations:**

Yes

**Strengths And Weaknesses:**

Strength:
-	Although masked inference is not a new idea, this work targets at a more practical inference speed optimization compared to existing dynamic-masking approaches. Several critical modifications for the real efficiency of this approach are adopted in the inference pipeline: fused operations, granularity choices of the mask etc.
-	Experiments with ResNet and RegNet validates the effectiveness of this method on several tasks: ImageNet classification and object detection. It reduces inference time by 23% and 45% on V100 and TX2 platform respectively without significant drop on accuracy.
Weaknesses:
-	The framework can only be applied to models with ResNet block whose major computations are located in the 3x3 conv of the network blocks. I don’t think it work well for other efficient models such as MobileNet and ShuffleNet that does not have heavy 3x3 convs.
-	Only GPU platforms are tested. Does this approach generalize to other types of hardware such as CPU?
-	For the speed comparison, what kind of cuda runtime environment are you using for the experiments? What are the cuda version, cudnn version? are you using TensorRT? These details are not given in the main text.

---

> ### Author Response · Authors · 2022-08-02
> **Response to Reviewer BmLK**
>
> Thanks for the positive comments. Our responses to your mentioned weaknesses are as follows:
>
>
>
> ### Weaknesses
>
> > The framework can only be applied to models with ResNet block whose major computations are located in the 3x3 conv of the network blocks. I don’t think it work well for other efficient models such as MobileNet and ShuffleNet that does not have heavy 3x3 convs.
>
> - We would like to clarify that the effectiveness of our method is **not affected by the shape of convolution**, because **our method makes use of the spatial redundancy in feature maps, rather than the structure redundancy in convolutional kernels**. Whatever the convolutional kernel size is, the redundant computation on less informative regions can be skipped. In other words, the computation of both 3x3 conv and 1x1 convs can be reduced during inference.
>
> - Actually, the **RegNets that we experimented on do not have heavy 3x3 convolutions**. We compute the FLOPs that each layer consumes in a RegNetY-800M, and the results show that the 3x3 convs only takes ~**20%** computation in most blocks (and the 1x1 convs take ~80%). Our results in Fig. 7 (b) validate that **our method works well on such efficient models without heavy 3x3 convs, and the LAS-RegNets outperform MobileNets in both theoretical and practical efficiency**.
>
> - We also tested our method on MobileNet-v2. When achieving similar accuracy (~72.1%) on ImageNet, **our conclusion that coarse-grained spatially adaptive inference (S>1) achieves better efficiency than the previous methods (S=1) still holds**. For example, the inference latency of our LAS-MobileNet-v2 with S=8-4-7-1 and S=1-1-1-1 are 5ms and 7ms respectively (the former is 28.5% faster) on TX2 GPU.
>
>
>
> > Only GPU platforms are tested. Does this approach generalize to other types of hardware such as CPU?
>
> - We would like to point out that it is has been validated by many previous works (references [7,27,31] in our paper) that the realistic speedup of spatially adaptive computation on CPUs **can already be achieved**, because their is a strong correlation between latency and FLOPs on CPUs. In contrast, **the realistic speedup on GPUs remains a more challenging problem and is rarely explored by researchers**. Therefore, we mainly focus on the GPU platform in this paper. We have updated the introduction section (line 41-43) to clarify this.
> - Of course, our models can achieve realistic speedup on CPUs. For example, we have conducted speed tests on Intel(R) Xeon(R) CPU (E5-2698 v4 @ 2.20GHz). The results suggest that the **practical speedup ratio of our coarse-grained spatially adaptive computation (LASNets with S>=2) generally matches the theoretical speedup ratio**. In contrast, the speedup ratio of the finest granularity (S=1) lags behind the theoretical results by 30%~60%. More comprehensive empirical study will be included in the paper.
>
>
>
> > For the speed comparison, what kind of cuda runtime environment are you using for the experiments? What are the cuda version, cudnn version? are you using TensorRT? These details are not given in the main text.
>
> - The inference code is implemented in C++/CUDA. The CUDA version is 11.6. Because **cuDNN and TensorRT have not supported some operations required by dynamic inference** (patch-level scatter-add and gather-conv3x3 operations), we implement the dynamic kernels without cuDNN or tensorRT. The compared speed of the baseline (static) models are also tested in our own implementation. According to our experiments, **our implementation of static operators runs ~16% faster than cuDNN-implemented static operators.** Therefore, the advantage of our dynamic operators over the cuDNN-implemented static operators is more significant.
> - These details have been included in our updated supplementary material, and we will release the code upon acceptance.

---

> > ### Comment · Reviewer_BmLK · 2022-08-07
> > **Updated comments**
> >
> > The response addressed most of my concerns. I will raise my rating if no major issue is found by other reviewers.

---

> > > ### Author Response · Authors · 2022-08-08
> > > **Response to the updated comments by BmLK**
> > >
> > > Thanks for your appreciation!

---

### Official Review · Reviewer_3HuP · 2022-07-13

**Rating:** 7
**Confidence:** 4
**Soundness:** 3 good
**Presentation:** 3 good
**Contribution:** 4 excellent

**Summary:**

Paper tackles the problem of dynamic inference where the forward graph will depend on the input data. It focuses on real speedups instead  of theoretical one, this makes the work more valuable. There is a latency prediction model that can estimate the latency by considering the algorithms, scheduling and hardware properties. Experiments are performed on image classification and demonstrate latency reduction of 23 to 45% depending on the hardware superiority.

**Questions:**

- How does hardware parameter vector H look like?
- The activation factor $r$, is it preselected or computed automatically? What would be con\pros of both solutions?
- What is the architecture of masker?
- Is the mask the same for all input/output channels? Or it does vary?
- Why the case of S=56 is not considered? If my understanding is correct then the entire feature map will be removed which is helpful to reduce latency.
- Explain how sparsity is set of different layers and why S is different for different models and layers in section 4.3.1
- Implementation of the 2 branch model in Figure 2 is not clear. If the mask is adaptive then there might be more than 1 candidate with sparse convolution, how would Gumbel softmax sample them? How M is regularized?
- How is latency computed in Figure 7? Was the setup the same (operator fusion etc) for all models considered? Those questions raise because skipping the entire block as in Convolutional-AIG should be more efficient in terms of latency reduction.
- Where was the framework implemented? If it was C++ then comparing with cuDNN and TesnorRT would be valuable as there is additional model optimization.
- Will the code be released? The latency prediction model might be useful to community for future research.

Minor:

- In the first section, the coarse granularity is mentioned. It would be helpful to explain it once it appears in the text.

**Ethics Review Area:**

["I don’t know"]

**Limitations:**

- Limitations are not listed by authors
- There are seem to be a possible a set of hyper-parameters that need to be tuned like S etc.
- Some details are missing and clarifying them will be helpful.
- It is not clear how different convolution implementations (like Winograd) will benefit from the work.


**Strengths And Weaknesses:**

Strengths:

+ Dynamic inference is a field of great interest. Naturally, people spent different effort to perform vision tasks so should deep network.
+ The method considers the real latency and not FLOPs. Additionally, demonstrating real speed-up is greatly appreciated.
+ Multiple paths are considered during training, and a single path is executed during inference.
+ Pare is well written and the content comes smoothly.

Weaknesses:

- Using distillation during fine-tuning will increase accuracy and might be not fair comparing to the original model.
- When compared to the previous work like Conv-AIG, the training recipes might be different (authors start from trained model and do 100 more epochs). The over training pipeline is not described as well, things like augmentations, learning rate scheduler etc should be mentioned. If those are not the same (as number of epochs) then authors should perform a fait comparison with those being the same.
- There are questions in the section before that are not clear. Understanding them will help to evaluate paper better.

---

> ### Author Response · Authors · 2022-08-02
> **Response to Reviewer 3HuP (part3)**
>
> > How is latency computed in Figure 7? Was the setup the same (operator fusion etc) for all models considered? Those questions raise because skipping the entire block as in Convolutional-AIG should be more efficient in terms of latency reduction.
>
> - The overall latency is obtained by **summing up the latency of all the blocks in a network**. The latency of each block is estimated using the latency prediction model by considering the latency of both data movement and computation. More detailed description of our latency prediction model is included in the updated paper and supplementary material.
> - **The setup of operator fusion is decided based on the averaged sparsity of a block.** For example, when the sparsity is high (very few pixels are selected), the latency bottleneck would be memory access rather than computation. In this situation, it would be more efficient to conduct operator fusion. We calculate the averaged sparsity of each block on the ImageNet validation set and decide whether to fuse some operations. This is practical thanks to our proposed latency prediction model, which helps us to efficiently analyze the latency bottleneck.
> - Although skipping the entire block as in Conv-AIG (or our coarsest granularity S=56-28-14-7) is easier to implement for fast inference, it might degrade the network performance (please also refer to our response to Question 5). Note that in our experiments for the variant of S=56-28-14-7 (which is similar to Conv-AIG), the operator fusion is considered in the same way as other granularity settings.
>
>
>
> > Where was the framework implemented? If it was C++ then comparing with cuDNN and TesnorRT would be valuable as there is additional model optimization.
>
> - The latency predictor is implemented in Python and the inference code is implemented in C++/CUDA. Because **cuDNN and TensorRT have not supported the dynamic operators in our method**, we can only conduct comparisons in our framework. Our results have shown that **the implemented dynamic operators run faster than the static operators**.
> - We have also compared the static operators implemented in our framework with cuDNN. The results show that **our implementation of static operators is also more efficient than the cuDNN library**. For example, our implementation of a 3x3 convolution layer in the first stage of a ResNet runs faster than a cuDNN-implemented layer by ~16%.
> - Based on the above analysis, the conclusion is that **the dynamic operators (our implementation) outperform the static operators (our implementation), and the later is faster than cuDNN-implemented static operations**. Therefore, the advantage of our dynamic operators over the cuDNN-implemented static operators is actually more significant.
>
>
>
> > Will the code be released? The latency prediction model might be useful to community for future research.
>
> - Thanks for your appreciation. We will release the code upon acceptance.
>
>
>
> > In the first section, the coarse granularity is mentioned. It would be helpful to explain it once it appears in the text.
>
> - Thanks, we have updated the paper (line 57-59) to explain this.
>
>
>
> ### Limitations
>
> > Limitations are not listed by authors
>
> - Actually, we discussed the limitation at the end of the paper. A more detailed analysis is included in the updated supplementary material.
>
>
>
> > There are seem to be a possible a set of hyper-parameters that need to be tuned like S etc.
>
> - As mentioned in our response to Question 6, **S is decided  based on our latency analysis experiments in Section 4.2. This can be achieved thanks to our proposed latency prediction model.** Other hyper-parameters are quickly decided in our early experiments.
>
>
>
> > Some details are missing and clarifying them will be helpful.
>
> - Thanks, we have updated the paper and the supplementary to include more details. The presentation will be further polished.
>
>
>
> > It is not clear how different convolution implementations (like Winograd) will benefit from the work.
>
> - Winograd can benefit from our coarse-grained dynamic inference. For 3x3 convolution, we can use the Winograd algorithm when the input of a patch has 4x4 or more pixels (S>=2). Because the normal implementation of convolution is the most commonly used, we now only experiment on the normal implementation of convolution. Acceleration in other implementations is one of the interesting research directions which could be explored in the future.

---

> > ### Comment · Reviewer_3HuP · 2022-08-08
> > **Response**
> >
> > Thanks authors for providing detailed answers. I will tend to keep my initial score. It will be critical to release the code as I think community will benefit from latency predictor even for full layers with no spatial adaptivity.

---

> > > ### Author Response · Authors · 2022-08-09
> > > **Response to the updated comments by 3HuP**
> > >
> > > Thanks for your appreciation, we will polish the code and release it upon acceptance.

---

> ### Author Response · Authors · 2022-08-02
> **Response to Reviewer 3HuP (part2)**
>
>
>
> > The activation factor r, is it preselected or computed automatically? What would be con\pros of both solutions?
>
> - The activation rate r is **computed automatically based on the output of the masker**. It cannot be preselected, because we want the maskers to **adaptively** decide which pixels on a feature map to be calculated in different blocks. The calculated r determines the FLOPs of the block, and then the overall FLOPs of a network can be obtained. We add a regularization item in the loss function to optimize the overall FLOPs of our networks to a certain target (Sec. 3.5, line 203 of the paper). By doing so, the maskers will learn to adaptively select the most important regions in feature maps during inference (please also refer to our visualization results in Fig. 9 of the paper).
>
>
>
> > What is the architecture of masker?
>
> - **As described in the paper (line 138-139), the masker is composed of a pooling layer and a conv1x1 layer.** Experimental results show that the architecture design of the makers is not crucial to accuracy, but a complex design would result in inefficiency without significant accuracy gains. Therefore, we finally adopt this design for faster inference.
>
>
>
> > Is the mask the same for all input/output channels? Or it does vary?
>
> -  **Yes, the mask is the same for all channels.** We did consider masking different regions for different channels. Empirical studies suggest that such an "overly" flexible operator would increase the difficulty of searching for the best scheduling strategy to achieve realistic speedup. Therefore, we keep the mask the same for different channels and only explore the granularity of spatially adaptive inference in this paper.
>
>
>
> > Why the case of S=56 is not considered? If my understanding is correct then the entire feature map will be removed which is helpful to reduce latency.
>
> - We omitted the detailed discussion for S=56-28-14-7 (the coarsest granularity) in the paper because spatially adaptive inference will reduce into a layer skipping operation in this situation. However, the main purpose of this paper is to explore the practical efficiency of the former paradigm.
> - We empirically compare our models with the mentioned variant (with similar computational cost). It is observed that although being slightly faster (13ms) than our LAS-ResNet-50 (S=4-4-2-1, 16ms), the accuracy could be significantly degraded from **76.6%** (ours, S=4-4-2-1) to **76.1%** (S=56-28-14-7).
> - We have also observed that the layer skipping scheme (S=56-28-14-7) degrades the performance on downstream tasks such as COCO instance segmentation. When achieving similar ImageNet accuracy (78.8%), LAS-ResNet101 (S=4-4-2-1, AP^box: **41.0%**, AP^mask: **37.0%**) significantly outperforms the layer skipping variant (S=56-28-14-7, AP^box: **40.4%**, AP^mask: **36.5%**).
> - Based on the above analysis, we conjecture that **the extreme situation of dropping an entire feature map may be too aggressive. Therefore, we mainly explore the granularity settings in the spatial-wise dynamic computation paradigm.**
>
>
>
> > Explain how sparsity is set of different layers and why S is different for different models and layers in section 4.3.1
>
> - As mentioned in our response to Question 2, **the sparsity is automatically calculated** based on the output of the maskers (please also refer to Sec 3.1, line 116 of our paper). We use a **regularization item in the loss function** (line 203 of the paper) to **control the overall computational cost of a network**, which leads to different sparsity in different layers.
> - As we describe in Section 4.2 (line 244-250), the **speedup ratios of different S settings are analyzed based on our latency prediction model**. We plot the relationship between speedup ratio and S for different blocks in Fig 6 of the paper, and a proper S is chosen for different blocks and models to achieve a better trade-off between latency and flexibility.
>
>
>
> > Implementation of the 2 branch model in Figure 2 is not clear. If the mask is adaptive then there might be more than 1 candidate with sparse convolution, how would Gumbel softmax sample them? How M is regularized?
>
> - **We have updated the presentation of Fig.2 in the paper**. In fact, our model is not a two-branch architecture. The two "branches" in the original Fig. 2 denote the training-time masking scheme and the test-time operator fusion, respectively.
> - In our case, **each element of a mask makes a binary decision** on whether to allocate computation to a pixel/patch on the feature map. A masker takes the input feature as input and adaptively decides which locations should be computed. Gumbel Softmax is a commonly used technique that facilitates the end-to-end training of such "discrete" decisions.
> - As we mentioned above, a regularization item is used in the loss function to control the overall computational cost of a network.

---

> ### Author Response · Authors · 2022-08-02
> **Response to Reviewer 3HuP (part1)**
>
> Thanks for your detailed comments. The mentioned questions are addressed as follows.
>
> ### Weaknesses
>
> > Knowledge distillation.
>
> - First, we would like to clarify that we obtained the baseline results from our reference [27] (dynConv). Moreover, the dynConv method could be seen as one variant (S=1-1-1-1, which is trained with exactly same strategy with other S settings) in our framework. Therefore, we believe that the comparison (especially among our models with different granularity settings) in terms of the trade-off between accuracy and efficiency is fair.
> - Second, we have validated that under the same setting, our models are still more efficient while achieving comparable accuracy.  We finetuned the original pre-trained ResNet-101 with the same teacher for distillation. After finetuned for 100 epochs, the original ResNet-101 achieves **78.9%** Top-1 accuracy while consuming ~**7.8G** FLOPs (**38ms** on TX2). In contrast, our dynamic model (S=4-4-2-1) achieves **78.8%** accuracy while consuming ~**5.6G** FLOPs (**28ms** on TX2). It can be observed that even with the same training strategy, our dynamic models can significantly improve the inference efficiency of the original model when yielding comparable accuracy.
>
>
>
> > Training recipe.
>
> - Actually, the training pipelines of spatial-wise dynamic models adopted by previous works vary a lot. There are mainly two lines: one is the pretrain-finetune paradigm adopted in SACT [5] and dynConv [27], and the other trains the models from scratch, but for more (200) epochs for better convergence [31, 7]. As our model architectures are most similar to those in dynConv [27], **we follow the training setup in [27]** to avoid the "dead residual problem". We conjecture that the training of spatial-wise dynamic models is generally more difficult than the layer skipping scheme in Conv-AIG, since the networks are required to make more complex decisions.
> - We use the standard data augmentation (normalization, RandomResizedCrop, and RandomHorizontalFlip) which is the same as most compared methods, and adopt the cosine-shape decaying scheduler for the learning rate. We have updated the supplementary material to clarify this.
> - Finally, we would like to emphasize that **all our models with different "granularity" settings are trained with the same recipe**. Therefore, the analysis in the experiment section and **our main conclusion that a properly "coarse granularity" (S>1) leads to better efficiency on GPUs is not affected by the training strategy**.
>
>
>
> > There are questions in the section before that are not clear. Understanding them will help to evaluate paper better.
>
> - We have updated the supplementary material (Section B.2) to include the aforementioned training details.
>
>
>
> ### Questions
>
> > How does hardware parameter vector H look like?
>
> - The considered **hardware properties** include the number of processing engines (PE), the off-chip memory bandwidth, the on-chip global memory bandwidth, and the throughput of FP32 computation of a PE. We omitted these details in the main text due to the page limit. **We have updated the paper (Section 3.3) and the supplementary material to include a more detailed description of our latency prediction model.**

---

### Author Response · Authors · 2022-08-02
**General Response to All Reviewers**

We thank all reviewers for their valuable comments! We are encouraged that the reviewers appreciate that:

1. Our work is well-motivated, which tries to solve a valuable and practical problem (all reviewers);
2. The proposed latency prediction model will be useful to the community (Reviewer 3HuP);
3. Our experimental results validate the effectiveness of our method (Reviewer BmLK), and the empirical analysis is insightful (Reviewer MLaG).

We have addressed the raised concerns

1. A more detailed description of our latency prediction model is provided in the paper (Section 3.3) and the supplementary material;
2. The detailed experimental settings and the speed testing environments are presented in the supplementary material;
3. Experiment results on the instance segmentation task are included in the supplementary material;
4. The language and the presentation of Figure 2 are improved.

Next, we address each reviewer's detailed concerns and questions point by point. We hope we have addressed all your concerns. Discussions are always open. Thank you!

---

### Meta-Review · Area_Chair_FRW2 · 2022-08-28

**Recommendation:** Accept
**Confidence:** Certain

**Metareview:**

The paper proposes latency-aware spatial-wise dynamic neural networks under the guidance of a latency prediction mode. reviewers arrived at a consensus to accept the paper.

**Award:**

No

---

### Decision · Program_Chairs · 2022-09-14

Accept